# Diet influences community dynamics following vaginal group B streptococcus colonization

Christina J. Megli,[1,2,3,4] Allison E. DePuyt,[4,5] Julie P. Goff,[1,4,6] Sarah K. Munyoki,[1,4,6] Thomas A. Hooven,[4,7,8,9] Eldin Jašarević[1,4,6]

**ABSTRACT** The vaginal microbiota plays a pivotal role in reproductive, sexual, and perinatal health and disease. Unlike the well-established connections between diet, metabolism, and the intestinal microbiota, parallel mechanisms influencing the vaginal microbiota and pathogen colonization remain overlooked. In this study, we combine a mouse model of *Streptococcus agalactiae* strain COH1 [group B *Streptococcus* (GBS)] vaginal colonization with a mouse model of pubertal-onset obesity to assess diet as a determinant of vaginal microbiota composition and its role in colonization resistance. We leveraged culture-dependent assessment of GBS clearance and culture-independent, sequencing-based reconstruction of the vaginal microbiota in relation to diet, obesity, glucose tolerance, and microbial dynamics across time scales. Our findings demonstrate that excessive body weight gain and glucose intolerance are not associated with vaginal GBS density or timing of clearance. Diets high in fat and low in soluble fiber are associated with vaginal GBS persistence, and changes in vaginal microbiota structure and composition due to diet contribute to GBS clearance patterns in nonpregnant mice. These findings underscore a critical need for studies on diet as a key determinant of vaginal microbiota composition and its relevance to reproductive and perinatal outcomes.

**IMPORTANCE** This work sheds light on diet as a key determinant influencing the composition of vaginal microbiota and its involvement in group B *Streptococcus* (GBS) colonization in a mouse model. This study shows that mice fed diets with different nutritional composition display differences in GBS density and timing of clearance in the female reproductive tract. These findings are particularly significant given clear links between GBS and adverse reproductive and neonatal outcomes, advancing our understanding by identifying critical connections between dietary components, factors originating from the intestinal tract, vaginal microbiota, and reproductive outcomes.

**KEYWORDS** diet–host–microbe interactions, vaginal microbiota, *Streptococcus agalactiae*, GBS

The composition of the vaginal microbiota has a profound impact on women's health. These microbial communities influence susceptibility to various medical conditions, including bacterial vaginosis (1–5), sexually transmitted infections (6), human immunodeficiency virus (7, 8), dyspareunia (9), vulvodynia (10), recurrent urinary tract infections (11, 12), and drug-resistant vulvovaginal candidiasis (13, 14). Furthermore, changes in the vaginal microbiota are also associated with an increased prevalence of adverse neonatal outcomes, such as spontaneous preterm birth (15–23), stillbirth (24–26), early-onset sepsis (27–30), and enduring health issues in surviving offspring (31). Despite the importance of the vaginal microbiota in human health, the determinants that regulate community structure, diversity, composition, and function across the lifespan remain incompletely understood.

Address correspondence to Eldin Jašarević, eldin.jasarevic@pitt.edu.

The authors declare no conflict of interest.

See the funding table on p. 15.

The female reproductive tract experiences continuous colonization by asymptomatic microbiota, some of which can transition from commensal members of the community to pathogens that may increase risk for negative health consequences. Among these vaginal pathobionts, *Streptococcus agalactiae*, or group B *Streptococcus* (GBS), are among the most clinically significant (32–34). The global prevalence of GBS vaginal colonization ranges from 20% to 40%, with an estimated 20 million infants exposed to maternal GBS around the time of delivery (35, 36). While typically asymptomatic in non-pregnant adults, GBS colonizing the vagina can ascend the reproductive tract and cause intra-amniotic infections during pregnancy (35, 37–39). In the absence of intrapartum prophylactic antibiotic therapy, these infections increase the risk of maternal systemic infections, stillbirth, preterm delivery, neonatal sepsis, meningitis, and enduring health challenges for infants (36, 40).

Emerging evidence points to the microbiota as a significant factor in promoting GBS colonization and persistence in the female reproductive tract (41). For instance, prolonged vaginal inflammation is associated with more frequent GBS colonization (42). Distinct vaginal community states that are characterized by the presence of inflammation-associated microbiota, including *Candida*, *Enterococcus*, and *Staphylococcus*, is associated with GBS colonization. In parallel, more recent efforts have established a mouse model of GBS vaginal colonization mirroring key clinical observations in humans (43, 44). In non-pregnant mice, approximately 75% of animals receiving a vaginal inoculation of GBS establish and sustain colonization for at least 48 h (43). Serial vaginal swab tests reveal that most mice spontaneously clear wild-type GBS within approximately a month after colonization. However, the timing of clearance varies significantly among individually housed mice (45), and the factors influencing time-to-clearance are unexplored. In contrast to nonpregnant female mice, GBS colonization around mid-pregnancy results in ascending chorioamnionitis, leading to intrauterine fetal demise and preterm delivery in this mouse model. Importantly, not all females acquire vaginal colonization, and some develop vaginal colonization without experiencing ascending chorioamnionitis (43). The intrinsic or extrinsic factors shaping these individual differences and heterogeneity in outcomes is unclear and warrants further study. Among intrinsic factors, the influence of systemic hormonal shifts and metabolic dysfunction on microbial communities and associated vaginal pathologies is well-recognized (46–51).

One currently unexplored extrinsic factor shaping vaginal microbiota is food. Diet is the most potent environmental factor determining individual variation in microbiota structure, composition, and function (52–57). Studies exploring the role of diet on host–microbiota interactions have primarily focused on the intestinal tract. Evidence from these studies indicates that intake of complex carbohydrates and saturated fats impact resistance or susceptibility to various pathogenic bacteria, trigger inflammation, and increase vulnerability to enduring poor health outcomes in the host (58–61). For instance, preconception obesity has been found to be associated with an increased risk of rectovaginal group B streptococcus (GBS) colonization in multiple studies (62, 63). A retrospective cohort study of women with singleton-term pregnancies showed that obese pregnant women were significantly more likely to be colonized by GBS compared to nonobese controls, even after adjusting for other factors such as race, parity, smoking, and diabetes. The prevalence of GBS colonization in the entire cohort was relatively high (25.8%), and obese gravidas had a 35% higher likelihood of testing positive for GBS compared to nonobese women (64). Another study also supported the association between obesity and GBS colonization, indicating that maternal obesity is a significant risk factor for GBS colonization (62). Chronic inflammatory changes and differences in intestinal microbiota in individuals with obesity have been suggested as potential contributors to the increased risk of GBS colonization (65).

Building on this extensive literature, we hypothesize that dietary composition influences vaginal GBS colonization dynamics. To gain deeper insights into how GBS persistence in the female reproductive tract might be influenced by diet, we integrated

an established mouse model of vaginal GBS colonization with our established model of pubertal-onset obesity. In this model, pubertal females consume a refined high-fat, low-fiber diet, and this leads to excessive weight gain and poor glucose tolerance, with endophenotypes consistent with metabolic syndrome in adulthood. Our approach involved investigating culture-dependent GBS density and clearance timing alongside culture-independent, sequencing-based reconstruction of the vaginal microbiota concerning diet, obesity, glucose tolerance, and microbial dynamics over time. Moreover, we determined that both dietary factors and GBS persistence in the mouse model are associated with the heightened presence of known pathogens of the female reproductive tract, such as *Enterococcus*. Through these studies, we are able to show a link between obesogenic diet and rectovaginal bacterial persistence in an animal model.

## RESULTS

### Body weight and glucose tolerance is diet-specific in female mice

We first validated diet effects on body composition using a model for obesity and glucose intolerance that starts during puberty in female C57BL/6N-TAC mice (66) (Fig. 1A). Initially, all mice were fed a standard Chow diet from birth until weaning. At weaning, on postnatal day 28, the mice were randomly placed on one of three diets: a low-fat, low-soluble fiber diet (rLFD, Research Diets D12450J); a high-fat, low-soluble fiber diet (rHFD, Research Diets D12492); or remained on the Chow diet (Fig. 1A and B). We have previously demonstrated that 10%–20% of the mice consuming the rHFD are resistant to weight gain and glucose tolerance; thus, we intentionally had this group have twice as many animals in anticipation of an overlapping physiologic response to diet (62).

Upon switching to these diets, significant differences in body weight trajectories were observed, consistent with prior findings (66, 67). Our current analysis showed significant effects of both time and diet, as well as their interaction, indicating that diet-related weight changes accumulated over time [repeated measures analysis of variance (RM ANOVA), main effect of time: $F_{5,280} = 218.3$, $P < 0.0001$; main effect of diet: $F_{2,56} = 14.05$, $P < 0.0001$; interaction: $F_{10,280} = 11.47$, $P < 0.0001$]. Specifically, mice on the rHFD gained weight more rapidly than those on the Chow or rLFD diets, while weight gain trajectories for Chow and rLFD diets were similar (Fig. 1C). Six weeks after switching to the respective diets, we assessed the impact on glucose tolerance using an intraperitoneal glucose tolerance test. We found that changes in glucose tolerance were linked to time, diet, and their interaction (RM ANOVA, main effect of time: $F_{4,228} = 35.46$, $P < 0.0001$; diet effect: $F_{2,57} = 48.36$, $P < 0.0001$; interaction: $F_{8,228} = 20.48$, $P < 0.0001$). Mice on the rHFD showed a significant delay in glucose clearance compared to those on the Chow and rLFD diets, which had similar glucose clearance rates (Fig. 1D). Additionally, rHFD mice had higher glucose levels after the glucose tolerance test compared to the other groups, while Chow and rLFD mice showed no difference in glucose levels (Fig. 1D). We found a significant correlation between weight gain and glucose levels in the rHFD group, but not in the Chow or rLFD groups (rHFD correlation: $r^2 = 0.4004$, $P = 0.0155$; no significant correlations for Chow and rLFD) (Fig. 1F). These findings support existing research on how diets with different fat and soluble fiber contents affect weight gain and glucose tolerance in a diet-specific manner (53, 68–70).

### Vaginal GBS COH1 colonization and persistence is diet specific in female mice

Epidemiological studies have suggested increased prevalence in GBS colonization and persistence among obese women (64, 65, 71). Guided by this clinical data, we sought to determine the potential impact of diet-induced metabolic syndrome and obesity on GBS colonization and persistence. To achieve this, we utilized an established murine GBS vaginal colonization model (43). Adult non-pregnant female mice consuming Chow, rLFD, and rHFD diets were estrus synchronized and then colonized with $1 \times 10^7$ colony-forming units (CFU) of GBS COH1. Colonization, persistence, and clearance were assessed

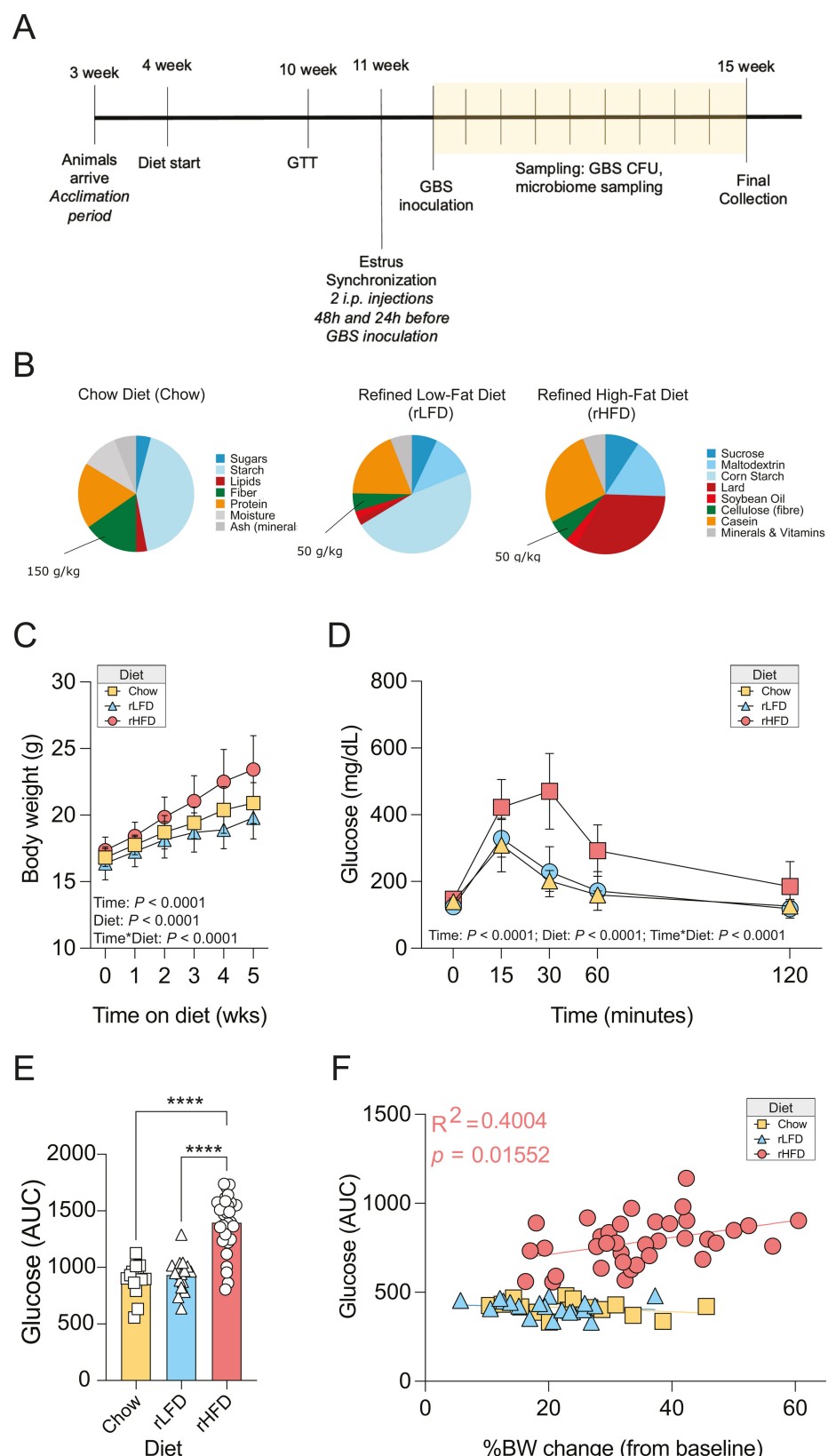

**FIG 1** Dietary compositions that vary in fiber and fat influence whole-body weight and glucose homeostasis. (A) Schematic of experimental design to evaluate the role of diet on murine vaginal colonization of *S. agalactiae* (GBS) COH1. Weaning age (P28) C57Bl/6N TAC female mice consuming a Chow diet were randomly switched to consume either Chow, an rLFD, or an

**FIG 1** (Continued)

rHFD for 6 weeks. We examined the influence of diet on whole-body metabolism by recording body weight measurements, and as adults (P70), females received an intraperitoneal glucose tolerance test to assess glucose homeostasis while consuming these diets. Females were then single housed, and following a 2 week recovery, females were hormonally synchronized with an encapsulated 17b-estradiol and, 24 h later, vaginally inoculated with $1 \times 10^7$ colony-forming units (CFU) of GBS COH1 or vehicle. Vaginal swabs were collected every 48 h for CFU enumeration and microbial profiling ( a total starting of $n = 13$ Chow, 16 rLFD, 30 rHFD female mice/timepoint, total $N = 590$ samples). (B) Nutritional composition of diet and ingredients for the Chow, rLFD, and rHFD. Refined diets isocaloric matched and the insoluble fiber cellulose as the primary source of dietary fiber, lacking soluble fiber. (C) Diet-specific effects on body weight trajectory. Body weight of females was significantly changed over time [repeated measures analysis of variance (RM ANOVA), main effect of time, $F_{5,280} = 218.3$, $P < 0.0001$], across diets (RM ANOVA, main effect of diet, $F_{2,56} = 14.05$, $P < 0.0001$), and their interaction (RM ANOVA, time * diet, $F_{10,280} = 11.47$, $P = <0.0001$). rHFD females showed accelerated weight gain compared with Chow ($t_{56} = 4.309$, $P = 0.0097$) and rLFD females ($t_{56} = 7.200$, $P < 0.0001$). Chow and rLFD females showed the same weight gain trajectory ($P = 0.293$). (D) Diet-specific effects on glucose tolerance. Glucose levels significantly changes over time (RM ANOVA, main effect of time, $F_{4,228} = 35.46$, $P < 0.0001$), across diets (RM ANOVA, main effect of diet, $F_{2,57} = 48.36$, $P < 0.0001$), and their interaction (RM ANOVA, time * diet, $F_{8,228} = 20.48$, $P = <0.0001$). rHFD shows significant delay in glucose clearance compared with Chow ($P < 0.0001$) and rLFD females ($P < 0.0001$). Chow and rLFD females show similar glucose clearance profiles ($P = 0.900$). (E) Diet-specific effects on circulating glucose levels (one-way ANOVA, main effect of treatment, $F_{2,57} = 41.33$, $P < 0.0001$). rHFD shows significantly higher circulating glucose levels than Chow ($t_{57} = 10.58$, $P < 0.0001$) and rLFD ($t_{57} = 10.29$, $P < 0.0001$). Chow and rLFD females show similar glucose levels in response to a glucose tolerance test ($P = 0.89$). (F) Significant correlation between body weight gain and glucose levels in rHFD females but not Chow and rLFD females ($r^2 = 0.4004$, $P = 0.0155$ for rHFD; all $Ps > 0.05$ for Chow and rHFD); $N = 13$ Chow, 16 rLFD, 30 rHFD female mice.

by quantifying CFU density from vaginal swabs collected every 2–3 days for 30 days. Quantification of CFU densities showed that initial colonization was not different between groups with 80%–90% of the groups with CFU detected at 48 h. Kaplan–Meier analysis revealed that rLFD and rHFD females demonstrated a prolonged time to colonization clearance of GBS COH1 compared with Chow females (Chow vs rHFD, $P = 0.032$; Chow vs rLFD, $P = 0.0041$, Log-rank test). However, rLFD and rHFD showed similar rates of colonization clearance ($P = 0.1568$, Log-rank test) (Fig. 2A). Additional analyses using linear regression applied to GBS COH1 densities over the course of a 30-day period revealed that GBS COH1 density decreased in Chow females ($r^2 = 0.049$, $P = 0.036$), increased in rLFD females ($r^2 = 0.115$, $P = 0.0011$), and remained constant in rHFD females ($r^2 = 0.006$, $P = 0.325$) (Fig. 2B).

We conducted a series of correlation analyses to examine the association between diet-induced body weight gain and GBS colonization in our preclinical model. We found no significant associations between diet condition, circulating glucose levels, and GBS COH1 clearance patterns (Fig. 2C). Similarly, we found no associations between diet condition, body weight gain, and GBS COH1 clearance patterns (Fig. 2D). This may suggest that diet-induced changes to body weight changes, glucose tolerance, and glucose levels are not correlated with vaginal GBS COH1 colonization clearance. Consistent with the work on diet–microbiota associations in the intestinal tract, our results suggest that consumption of diets of fat and soluble fiber that vary in proportion impacts GBS COH1 colonization clearance and persistence in this mouse model.

## Diet influences temporal dynamics of vaginal microbiota following GBS COH1 in female mice

We next determined whether diet condition and GBS colonization influenced vaginal microbiota composition. We calculated beta and alpha diversity measures and compared them in relation to colonization status, diet, and days post-inoculation (Fig. 3A through C). Permutational multivariate analysis of variance revealed a significant effect between Vehicle and GBS-colonized females [permutational multivariate analysis of variance (PERMANOVA), $F = 67.211$, $r^2 = 0.1786$, $P < 0.001$]. Vaginal community structure of females consuming refined diets showed a significant community structure relative to Chow diet females (PERMANOVA, $F = 7.6593$, $r^2 = 0.0473$, $P < 0.001$). Vaginal community structure

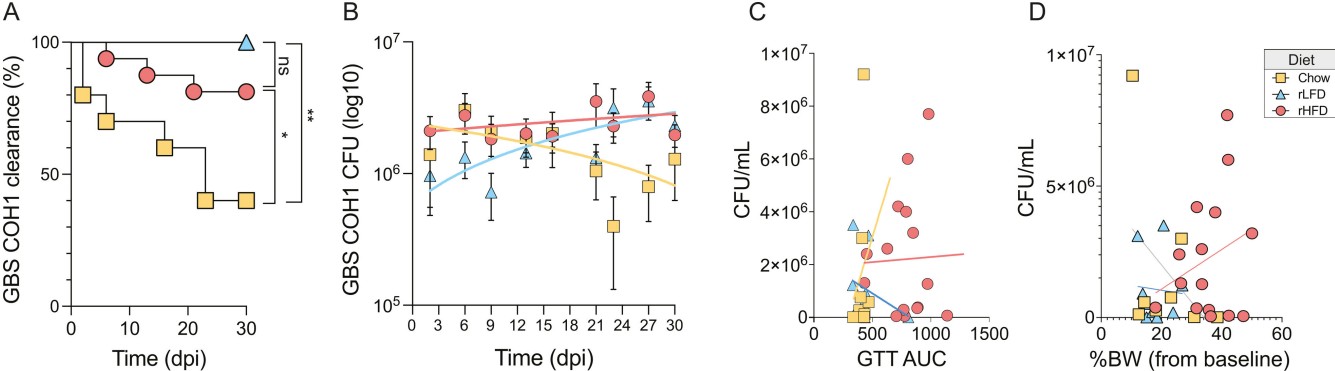

**FIG 2** Dietary compositions that vary in fiber and fat influence vaginal GBS COH1 colonization and persistence in the female mouse. (A) Diet influences GBS COH1 colonization persistence in a mouse model of vaginal colonization. Adult nonpregnant C57/BL6N TAC mice fed Chow, rLFD, or rHFD were vaginally colonized with $1 \times 10^7$ CFU GBS COH1. Kaplan–Meier curve representing colonization persistence, as determined by vaginal swabs every 2–3 days for 30 days, showing significant differences in clearance patterns between Chow and rHFD ($P = 0.032$, Log-rank test) as well Chow and rLFD females ($P = 0.0041$, Log-rank test). rLFD and rHFD showed similar rates of colonization persistence ($P = 0.1568$, Log-rank test). (B) Diet influences GBS COH1 CFU counts over a 30-day period. Over the course of a 30-day sampling period, linear regression analysis revealed that GBS COH1 CFU counts decreased in Chow females ($r^2 = 0.049$, $P = 0.036$), increased in rLFD females ($r^2 = 0.115$, $P = 0.0011$), and remained constant in rHFD females ($r^2 = 0.006$, $P = 0.325$). (C) No correlation between diet, circulating glucose levels, and GBS COH1 CFU at colonization. (D) No correlation between diet, body weight gain, and GBS COH1 CFU at colonization. Starting $N = 13$ Chow, 16 rLFD, 30 rHFD female mice, with sample size decreasing as females clear GBS.

changed significantly across the 30-day sampling period (PERMANOVA, $F = 5.7125$, $r^2 = 0.1314$, $P < 0.001$). These outcomes collectively indicate that both GBS colonization and dietary conditions are correlated with differences in vaginal microbial community structure.

GBS colonization, diet, and time were all noted to influence vaginal microbial diversity. We first compared between vehicle and GBS-colonized mice to determine if GBS colonization alters alpha diversity, which demonstrated that GBS colonization leads to reduced alpha diversity in GBS-colonized mice ($P < 0.0001$, Mann–Whitney test) (Fig. 3D). Further comparing Chow, rLFD, and rHFD female mice showed that diet is a significant factor influencing vaginal microbiota alpha diversity ($P = 0.002$, Kruskal–Wallis test). rLFD exhibited reduced alpha diversity compared to Chow ($P = 0.0488$, Mann–Whitney test) and rLFD compared with rHFD ($P = 0.0004$, Man–Whitney test) (Fig. 3E). Tracking Chow, rLFD, and rHFD females over the 30-day period indicated that GBS COH1 colonization affects vaginal microbiota alpha diversity (main effect of time, $F_{9,245} = 12.07$, $P < 0.0001$, mixed-effects model). The Shannon diversity index (SDI) metrics decreased from 0 to 6 days post-inoculation (dpi) with GBS COH1 across all dietary conditions (0 vs 6, $t_{245} = 4.963$, $P = 0.0188$), with community diversity failing to revert to baseline levels within the 30-day sampling period (0 vs 30 dpi, $t_{249} = 9.705$, $P < 0.0001$, Tukey *post hoc*) (Fig. 3F).

Extending our analysis to observed counts within the vaginal microbiota of Chow, rLFD, and rHFD female mice across 30 days, a significant time effect on vaginal microbiota alpha diversity emerged (main effect of time, $F_{9,245} = 49.76$, $P < 0.0001$, mixed-effects model), and this effect was further compounded by diet (time * diet interaction, $F_{18,249} = 2.202$, $P = 0.0039$, mixed-effects model). Observed amplicon sequence variant (ASV) counts decreased between 6 and 9 dpi with GBS COH1 similarly between all dietary groups (6 vs 9 dpi, $t_{249} = 14.61$, $P < 0.0001$, Tukey *post hoc*). Like the Shannon diversity index analysis, observed ASV counts did not recover to baseline levels within the 30-day sampling period (0 vs 30 dpi, $t_{249} = 9.984$, $P < 0.0001$, Tukey *post hoc*) (Fig. 3G). We then determined the effect of GBS colonization on vaginal community structure by comparing the relative abundance between vehicle and GBS-colonized mice (Fig. 3H), as expected introduction of GBS altered the community diversity and led to relative abundance of GBS. Comparison of vaginal microbiota composition between Chow, rLFD,

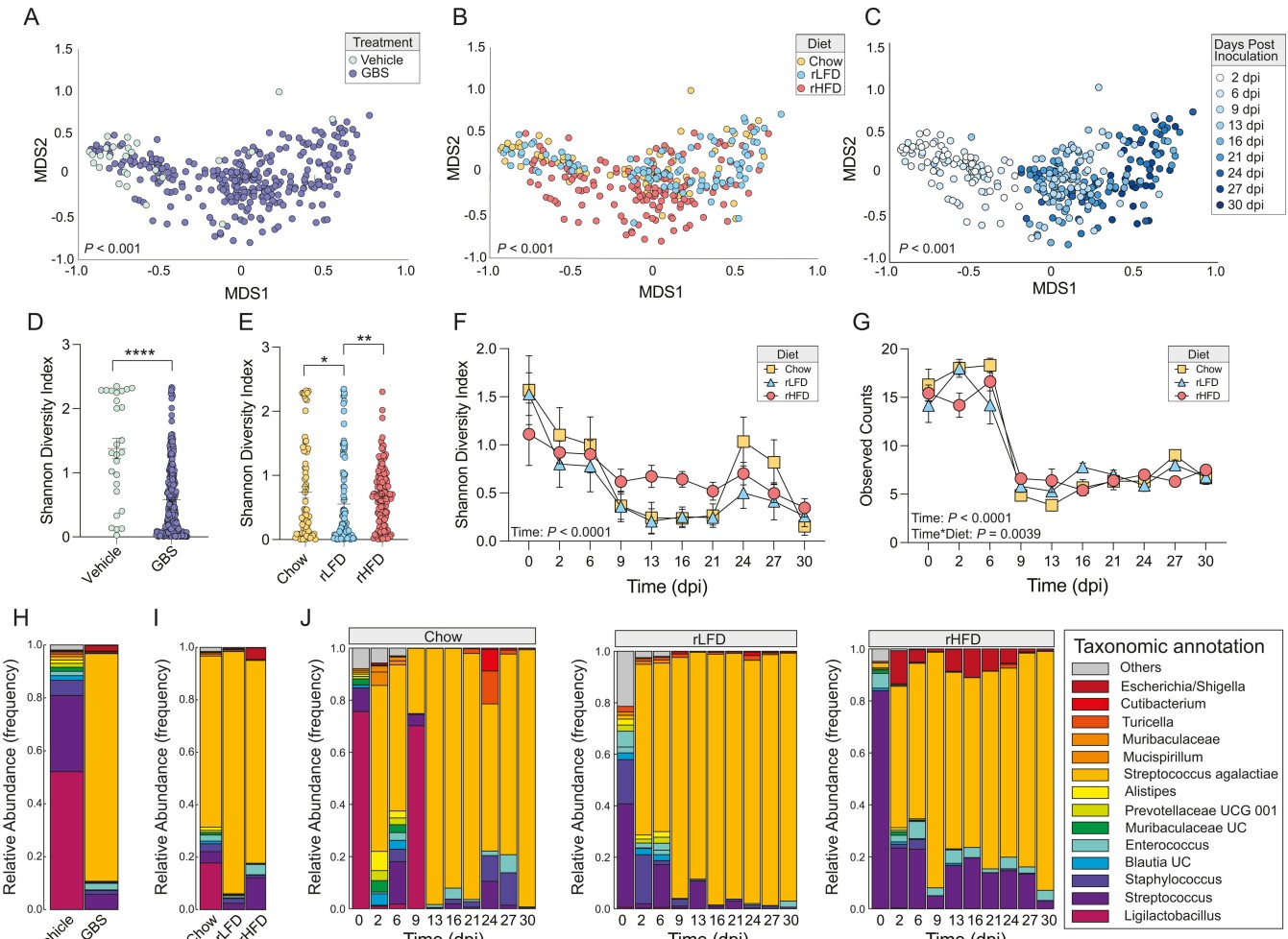

**FIG 3** Dietary compositions that vary in fiber and fat influence temporal dynamics of vaginal microbiota following GBS COH1 in the female mouse. (A–C) Non-metric multidimensional scaling analysis showing treatment (PERMANOVA, $F = 67.211$, $r^2 = 0.1786$, $P < 0.001$) (A), diet (PERMANOVA, $F = 7.6593$, $r^2 = 0.0473$, $P < 0.001$) (B), and sampling time effects (PERMANOVA, $F = 5.7125$, $r^2 = 0.1314$, $P < 0.001$) (C) on vaginal microbial community structure. Sampling time was every 2–3 days following colonization with $1 \times 10^7$ CFU GBS COH1. (D) Dot plot of Shannon diversity index (SDI) in the vaginal microbiota of vehicle only and GBS-colonized mice showing that vaginal microbiota alpha diversity is reduced in GBS-colonized females compared with Vehicle females ($P < 0.0001$, Mann–Whitney test). Data are represented as mean ± SEM. Table S1 provides all comparisons. (E) Dot plot of Shannon Diversity Index in the vaginal microbiota of Chow, rLFD, and rHFD female mice showing that vaginal microbiota alpha diversity is significantly altered by diet ($P = 0.002$, Kruskal–Wallis test). Alpha diversity is reduced in rLFD compared with Chow ($P = 0.0488$, Mann–Whitney test) and rLFD compared with rHFD ($P = 0.0004$, Mann–Whitney test). Data are represented as mean ± SEM. Table S1 provides all comparisons. (F) Longitudinal analysis of SDI in the vaginal microbiota of Chow, rLFD, and rHFD female mice over the course of a 30-day sampling period showing a significant effect of GBS COH1 colonization on alpha diversity in vaginal microbiota (mixed-effects model, main effect of time, $F_{9,245} = 12.07$, $P < 0.0001$). Shannon diversity index metrics decreased between 0 and 6 days post-inoculation (dpi) with GBS COH1 across all dietary conditions (0 vs 6, $t_{245} = 4.963$, $P = 0.0188$). Alpha diversity, as measured by SDI, failed to recover to baseline levels within the 30-day sampling period (Tukey *post hoc*, 0 vs 30 dpi, $t_{249} = 9.705$, $P < 0.0001$). (G) Longitudinal analysis of observed amplicon sequence variant (ASV) counts in the vaginal microbiota of Chow, rLFD, and rHFD female mice over the course of a 30-day sampling period showing a significant effect of time on vaginal microbiota alpha diversity (mixed-effects model, main effect of time, $F_{9,245} = 49.76$, $P < 0.0001$), and this effect was influenced by diet (mixed-effects model, time * diet interaction, $F_{18,249} = 2.202$, $P = 0.0039$). Observed counts decreased between 6 and 9 dpi with GBS COH1 similarly between all dietary groups (Tukey *post hoc*, 6 vs 9 dpi, $t_{249} = 14.61$, $P < 0.0001$). Alpha diversity, as measured by observed counts, failed to recover to baseline levels within the 30-day sampling period (Tukey *post hoc*, 0 vs 30 dpi, $t_{249} = 9.984$, $P < 0.0001$). (H) The average relative abundance of the top 15 vaginal taxa between females in either Vehicle or GBS COH1 condition, confirming increased relative abundance of reads mapping to *S. agalactiae* only in GBS-colonized females but not vehicle. (I) Average relative abundance of the top 15 vaginal taxa between Chow, rLFD, and rHFD females colonized with GBS COH1, confirming detection of reads mapping to *S. agalactiae*. (J) Diet-specific dynamics in the average relative abundance of the top 15 vaginal taxa between Chow (left), rLFD (middle), and rHFD (right) females colonized with GBS COH1 across the 30-day sampling period, demonstrating an influence of diet on temporal dynamics of vaginal microbiota following colonization with GBS COH1. Starting $N = 13$ Chow, 16 rLFD, 30 rHFD female mice, with sample size decreasing as females clear GBS.

and rHFD displayed significant shifts in response to GBS COH1 colonization suggesting that these are altered by diet (Fig. 3I and J).

Thus, to assess whether dietary conditions influence vaginal microbiota composition following GBS COH1 colonization, we applied the linear discriminant analysis effect size method for differential abundance analysis, utilizing a false discovery rate (FDR) cut-off of $q < 0.05$. This analysis identified six taxa with altered relative abundance due to GBS COH1 colonization across dietary conditions (Fig. 4A through F). As expected, the reads mapping to *S. agalactiae* showed a significant increase within 48 h following GBS COH1 colonization, regardless of diet (main effect of time, $F_{9,244} = 26.43$, $P < 0.0001$, mixed-effect models; 0 vs 2 dpi, $t_{244} = 13.59$, $P < 0.0001$, Tukey *post hoc*). As expected, the reads mapping to *S. agalactiae* showed a significant increase within 48 h following GBS COH1 colonization regardless of diet likely reflecting the effect of inoculation. Although it is important to note that the 16S rRNA marker is not specific for GBS COH1 strain. Regardless, GBS was not detected via CFU in Vehicle-inoculated controls and 16S marker gene profiling in vehicle-inoculated controls.

The expansion in relative abundance of *S. agalactiae* was accompanied by the reduction of other key taxa in a diet-dependent manner. In Chow females, *Ligilactobacillus* and *Streptococcus* dominated prior to GBS COH1 inoculation (Fig. 4B and C). At 2 dpi, *S. agalactiae* relative abundance surged, becoming dominant in the reproductive tract of Chow females (Fig. 4A). Between 16 and 21 dpi, Chow females showed a transient expansion in the relative abundance of *Streptococcus*, succeeded by an increased relative abundance of *Staphylococcus* between 21 and 24 dpi (Fig. 4F). Following this, a bloom in *S. agalactiae* relative abundance occurred, eventually dominating the reproductive tracts of Chow females unable to clear GBS COH1 via culture-dependent methods (see Fig. 2A). Among rLFD females, *Staphylococcus* and *Streptococcus* dominated prior to GBS COH1 inoculation (Fig. 4C and F). At 2 dpi, the relative abundance of *S. agalactiae* dominated the female reproductive tract until 24 dpi when these females showed a transient increase in the relative abundance of *Staphylococcus* (Fig. 4A). However, this elevation was replaced by *S. agalactiae*. The consistent high relative abundance of *S. Agalactiae* in rLFD females accounts for their inability to clear GBS COH1, as indicated by culture-dependent methods. Finally, *Enterococcus* and *Streptococcus* represented the dominant taxa in rHFD females prior to GBS COH1 inoculation (Fig. 4C and D). In contrast to Chow and rLFD females, GBS COH1 inoculation in rHFD females increased in the relative abundance of *Escherichia/Shigella* from 2 to 24 dpi. This resulted in a co-occurrence of *S. agalactiae*, *Enterococcus*, and *Escherichia/Shigella*, a microbial signature that was not observed in the Chow and rLFD females (Fig. 4A, D and E). The decline in *Enterococcus* relative abundance coincided with an increased relative abundance of *Streptococcus* and a subsequent replacement of *S. agalactiae* by 30 dpi (Fig. 4A and E).

## DISCUSSION

Collectively, these results shed light on the associations between diet composition, vaginal microbiota, and GBS COH1 colonization dynamics in a mouse model. We observed diet-specific shifts in key taxa and in the response of the microbial community to the expansion of *S. agalactiae* in the reproductive tracts of the mice. The mice in the Chow group responded to GBS inoculation with an increase in *Streptococcus* and *Staphylococcus*, and the microbial niche was dominated by GBS in mice that were unable to clear the pathogen. Conversely, mice in the rLFD group had a robust, dominant establishment of GBS and had difficulty clearing the bacterium. Further, rHFD females displayed a distinct co-occurrence of *S. agalactiae*, *Enterococcus*, and *Escherichia/Shigella*, with a subsequent replacement of *S. agalactiae* at 30 dpi. These results uncouple the dietary effects on the vaginal microbiota from body weight gain and glucose tolerance and may shed light on the role of dietary components' varying proportions of saturated fat and soluble fiber in GBS colonization persistence. Importantly, our results underscore

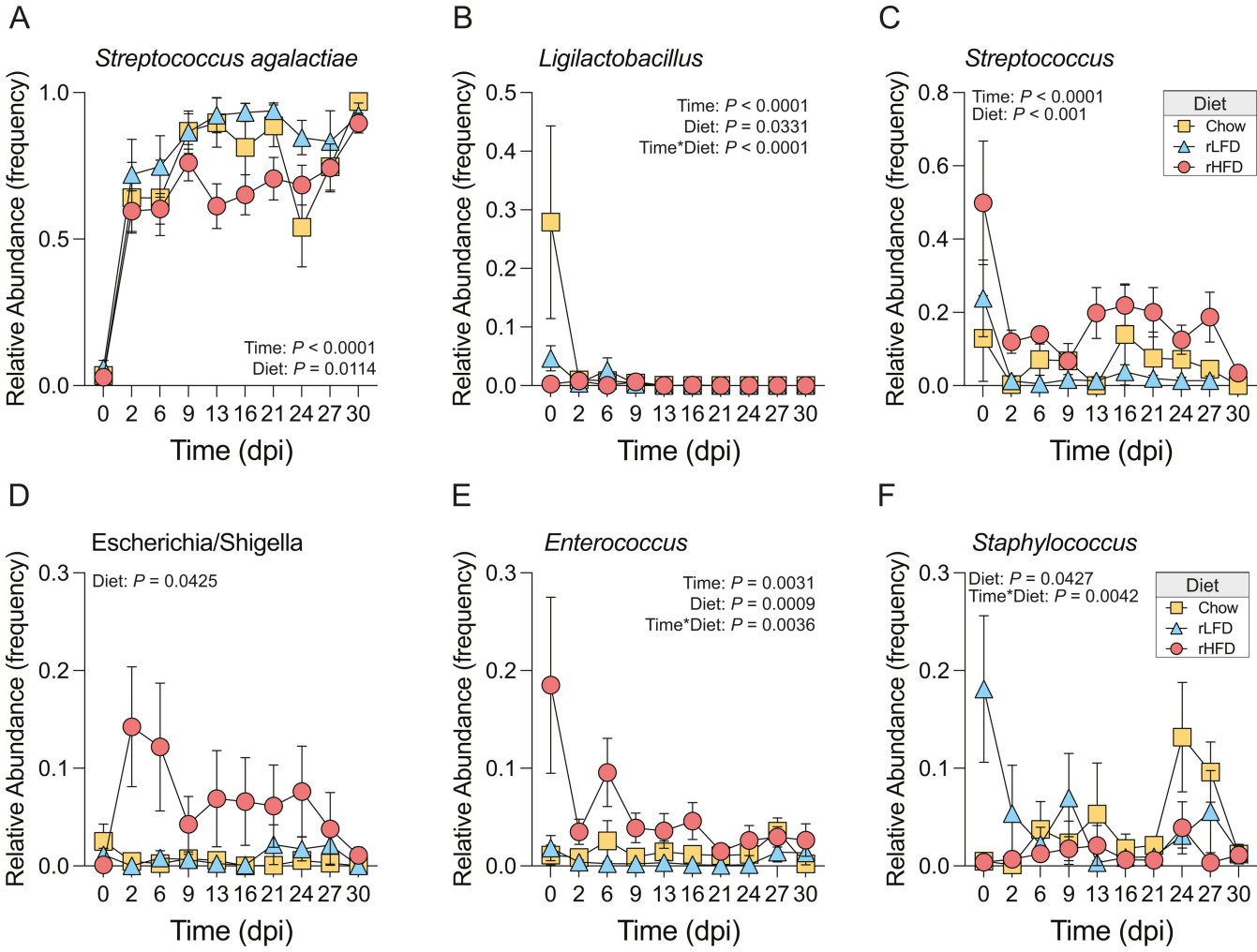

**FIG 4** Diet influences temporal dynamics of vaginal microbiota following colonization with GBS COH1. Analysis was conducted on the subcohort of animals that were GBS colonized, with 0 dpi reflecting a pre-inoculation timepoint. (A) Relative abundance of reads mapping to *S. agalactiae* in Chow, rLFD, and rHFD following vaginal colonization with GBS COH1. *S. agalactiae* relative abundance changed across the 30-day sampling period (mixed-effects model, main effect of time, $F_{9,244} = 26.43$, $P < 0.0001$), with a significant increase between 0 and 2 dpi (Tukey *post hoc*, 0 vs 2 dpi, $t_{244} = 13.59$, $P < 0.0001$). Diet showed a significant effect on *S. agalactiae* relative abundance (mixed-effects model, main effect of diet, $F_{2,36} = 5.076$, $P = 0.0114$), with relative abundance reduced in rHFD compared with rLFD females (Tukey *post hoc*, rLFD vs rHFD, $t_{36} = 4.496$, $P = 0.0083$). Table S2 provides all comparisons. (B) Relative abundance of reads mapping to *Ligilactobacillus* in Chow, rLFD, and rHFD following vaginal colonization with GBS COH1. *Ligilactobacillus* relative abundance changed across the 30-day sampling period (mixed-effects model, main effect of time, $F_{9,244} = 6.416$, $P < 0.0001$), differed across dietary conditions (mixed-effects model, main effect of diet, $F_{2,36} = 3.753$, $P = 0.0331$), and differed by diet across time (mixed-effects model, diet * time interaction, $F_{18,244} = 3.721$, $P < 0.0001$). *Post-hoc* analysis revealed that *Ligilactobacillus* relative abundance was increased in Chow females compared with rLFD and rHFD females, an effect that disappeared following colonization by GBS COH1 (all *P*s <0.05). (C) Relative abundance of reads mapping to *Streptococcus* in Chow, rLFD, and rHFD following vaginal colonization with GBS COH1. *Streptococcus* relative abundance changed across the 30-day sampling period (mixed-effects model, main effect of time, $F_{2,254} = 5.747$, $P < 0.0001$) and differed across dietary conditions (mixed-effects model, main effect of diet, $F_{2,36} = 9.490$, $P < 0.001$). *Streptococcus* relative abundance was higher in Chow and rLFD females compared with rHFD females (Tukey *post hoc*, Chow vs rHFD, $t_{36} = 4.323$, $P = 0.0114$; rLFD vs rHFD, $t_{36} = 5.708$, $P = 0.0008$). (D) Relative abundance of reads mapping to *Escherichia/Shigella* in Chow, rLFD, and rHFD following vaginal colonization with GBS COH1. *Escherichia/Shigella* relative abundance differed across dietary conditions (mixed-effects model, main effect of diet, $F_{2,36} = 3.451$, $P = 0.0425$). *Escherichia/Shigella* relative abundance trended higher in rHFD females compared with Chow females (Tukey *post hoc*, Chow vs rHFD, $t_{36} = 3.285$, $P = 0.0653$). (E) Relative abundance of reads mapping to *Enterococcus* in Chow, rLFD, and rHFD following vaginal colonization with GBS COH1. *Enterococcus* relative abundance changed across the 30-day sampling period (mixed-effects model, main effect of time, $F_{9,245} = 2.867$, $P = 0.0031$), differed across diets (mixed-effects model, main effect of diet, $F_{2,36} = 8.504$, $P = 0.0009$), and their interaction (mixed-effects model, diet * time interaction, $F_{18,245} = 2.221$, $P = 0.0036$). *Enterococcus* relative abundance was increased in rHFD females compared with Chow and rLFD females (Tukey *post hoc*, Chow vs rHFD, $t_{36} = 4.020$, $P = 0.0196$; rLFD vs rHFD, $t_{36} = 5.429$, $P = 0.0014$). (F) Relative abundance of reads mapping to *Staphylococcus* in Chow, rLFD, and rHFD following vaginal colonization with GBS COH1. *Staphylococcus* relative abundance (Continued on next page)

**FIG 4** (Continued)

differed across dietary conditions (mixed-effects model, main effect of diet, $F_{2,36} = 3.447$, $P = 0.0427$), and differed by diet across time (mixed-effects model, diet * time interaction, $F_{18,245} = 2.189$, $P = 0.0042$). *Staphylococcus* relative abundance differed between rHFD females compared with rLFD females (Tukey *post hoc*, rLFD vs rHFD, $t_{36} = 3.46$, $P = 0.0497$).

the significance of developing appropriate control diets for studies exploring diet-induced obesity, vaginal microbiota, and adverse outcomes associated with GBS colonization.

The microbial communities residing within the female reproductive tract have a significant impact on reproductive, sexual, and urological health. However, the role of food and diet composition in determining vaginal microbiota composition remains poorly understood (48, 49). This in contrast to the intestinal tract, where the links between diet, adiposity, glucose tolerance, and the intestinal microbiota have been extensively studied for over a decade (52, 53, 67–70, 72–77). Despite the potential for such research to provide new therapeutic and preventative approaches for female reproductive tract infections and related adverse health outcomes, similar mechanisms affecting the composition and function of the vaginal microbiota have received less attention. In this study, we used a mouse vaginal microbiota model in a two-by-three experimental design to assess vaginal microbiota composition dynamics in the presence or absence of experimental GBS colonization and under three different dietary conditions.

The vaginal microbiota composition of rodents significantly differs from that of humans. In humans, composition of the vaginal microbiota can be classified using community state types (CST), which are defined based on the proportions of the major taxa identified in the vaginal ecosystem (46, 47). Four of the five CST (CST I, II, III, and V) are dominated by *Lactobacillus* species, while the fifth CST (CST IV) is relatively *Lactobacillus* deplete, dominated instead by a heterogeneous group of strictly anaerobic bacteria that include *Prevotella*, *Gardnerella*, *Sneathia*, and other species (46, 47). Anaerobic bacterial abundance in the vagina is associated with bacterial vaginosis, which in turn correlates with susceptibility to sexually transmitted diseases including human immunodeficiency virus, chorioamnionitis, and premature birth (7, 17, 18, 21, 23, 46, 78–81). In contrast to humans, individual variation in mouse vaginal microbiota is determined by host genetics, strain, and vendor (81, 82). *Staphylococcus*, *Enterococcus*, and *Rodentibacter* species (formerly *Pasteurella pneumotropica*) typically dominate in the C57Bl/6 mouse vagina (82–85), while CD-1 mice have vaginal microbiota dominated by *Enterobacteriaceae* and *Proteus* species (86). Additionally, the current study used mice from Taconic, and these C57Bl/6 mice did not exhibit presence of vaginal *Rodentibacter*, compared to our previous work using C57Bl/6 mice from Jackson Laboratories where *Rodentibacter* was a dominant taxa (83, 85), pointing to important vendor differences in shaping rodent vaginal microbiota composition. Recent efforts have established a mouse CST (mCST) system analogous to the human classification, with five different mCST. Only one mCST (mCST IV) has a substantial proportion of *Lactobacillus* species, consistent with an extensive literature showing that a lack of lactobacilli is shared among rodents, ruminants, and non-human primates (87–90). While these species differences in vaginal microbiota composition caution against directly extrapolating preclinical findings to humans, they still provide valuable insights into vaginal microbiota and pathogen colonization dynamics. For instance, the colonization of GBS strain COH1 disrupts previously stable vaginal microbial communities, prompting a turnover and replacement with GBS. This shifts the vaginal ecosystem into a new community state where GBS becomes the dominant taxon (82).

Our current work adds key details to emerging narratives about intrinsic and extrinsic factors influencing vaginal microbiota dynamics. By introducing three diets to simultaneously housed mice, we compared the effects of altered fat and fiber intake, body weight, glucose tolerance, and circulating glucose levels on the mouse vaginal ecosystem. We hypothesized that the vaginal microbiota of rHFD females would be distinct from rLFD

mice. We also predicted that vaginal microbiota composition would overlap between Chow and rLFD females, given that the primary difference between these diets was the proportion of soluble fiber. In contrast to our predictions, the baseline vaginal microbiota of rLFD and rHFD female mice were more like each other than the Chow diet-fed mice. The vaginal microbiota of Chow-fed female mice was characterized by relatively high proportions of *Lactobacillus* (*Ligilactobacillus*), whereas the rLFD and rHFD mice were *Lactobacillus* deplete, colonized instead with high abundance of non-GBS *Streptococcus* (Fig. 3J and 4B).

Outcomes among the Chow-fed mice also diverged from rLFD and rHFD female mice upon GBS colonization. While all three cohorts developed high-density GBS colonization—which tended to overtake all non-GBS taxa, consistent with results from others (86, 87)—the Chow-fed mice cleared GBS vaginal colonization at higher rates than either of the refined diet groups. Given that the major distinguishing feature of the Chow and rLFD diets is the soluble fiber content, we hypothesize that fiber intake influences vaginal microbiota composition, favoring persistence of taxa that potentiate clearance of GBS.

Indeed, variation in the proportion of fat and fiber has been studied for its effects on the intestinal microbiota and found to influence local and peripheral metabolic and immune functions through modulation of the bacterial ecosystem (73, 91–96). Consumption of diets that are high in fat and low in fiber lacks nutrients that support commensal microbial communities. Loss of these communities opens a nutritional niche that renders increased availability of substrate necessary for pathogen outgrowth, resulting in an overall decrease resistance to colonization by pathogens and inability to clear the pathogen (97–99). Consistently, recent work demonstrated that gnotobiotic mice colonized with human intestinal microbiota and then fed a low-fiber diet demonstrated erosion of the colonic mucous barrier and susceptibility to colitis in a *Citrobacter rodentium* colonization model (73, 100). The notion that variations in the proportion dietary fat and fiber may influence the microbiota in the female reproductive tract is further supported by a recent study by Zou et al., which found that lactating dams deprived of dietary fiber raised juvenile mice with a tendency toward dysbiosis, obesity, and proinflammatory immunity (101). While the mechanism linking excess fat and fiber deprivation to offspring obesity and inflammation is not certain, the authors speculate that vertical transmission of maternal microbiota increases in proinflammatory Proteobacteria density, and an overall greater bacterial burden in the juvenile gut may underlie the effect (102, 103).

Our inability to detect significant associations between body weight, glucose tolerance, GBS clearance, and vaginal microbiota profiles may be related to several factors. First, the lack of association could arise from the nature of the endpoints being compared, including comparison of longitudinal dynamics (vaginal microbiota) with a single timepoint measure [glucose tolerance test (GTT)], or the measures used are not sufficiently sensitive to detect associations, such as with body weight. Second, the observed lack of association between metabolic parameters and vaginal microbiota may reflect recent findings in gut microbiota studies (67, 68, 73). For instance, recent work from our group and others showed that feeding mice diets either rLFD or rHFD led to significant changes in gut microbiota composition compared to a standard Chow diet (67, 68, 73). However, increases in body weight, body fat, and glucose intolerance were only observed in rHFD mice. Our data may suggest that a similar pattern may exist in the vaginal microbiota, indicating a potential dietary uncouple from microbiota effects and metabolic parameters. A rigorous examination of this hypothesis should be the focus of future research.

Our results also raise questions regarding the potential role of gut-derived signals influencing vaginal microbiota composition, susceptibility to pathogen colonization, and health outcomes. For instance, we found that varying the proportion of dietary fat and fiber influenced vaginal microbiota composition and GBS colonization dynamics. As changes to dietary fat and fiber can rapidly modify the metabolic and immunological output of the gut microbiota, the possibility that alterations in microbial metabolic

output may exert long-range effects on vaginal microbiota warrants further exploration. Our finding of changes in the vaginal microbiota resulting from excess fat and soluble fiber deprivation may indicate another linkage between the intestinal microbiota, the metabolites they produce, and long-ranging effects on the composition and function of bacterial ecosystems elsewhere in the body. Supporting this notion, a recent analysis of major metabolites revealed the presence of substrates like butyrate, erythritol, citalopram, and metformin in the cervicovaginal space, possibly originating from oral ingestion and modification by the intestinal tract (104). Uncovering novel mechanisms involved in gut-derived signals that regulate vaginal microbiota function could open new avenues for therapeutics and interventions aimed at improving adverse reproductive outcomes.

Furthermore, our study has some limitations. These include the inability to directly distinguish the effects of post-pubertal hormone signaling, altered immune signaling associated with obesity, vaginal tissue effects of obesity, and the impacts on immune signaling and tissue regulation in impaired glucose tolerance, as well as the mechanisms underlying bacterial colonization in the vagina. We hypothesize that the determinants of GBS colonization and persistence in this model are multifactorial. Future research should aim to elucidate the mechanistic links between these factors and the observed phenotypes.

Additional studies are now necessary to investigate the role of maternal diet in neonatal susceptibility to GBS intestinal colonization. This is particularly relevant as GBS colonization represents a key risk factor for early-onset sepsis, which has been modeled in newborn mice (105). Furthermore, our findings suggest that associations between diet, vaginal microbiota, and GBS colonization dynamics have practical implications for studying diet–host–microbe interactions in preclinical settings. Differences in rodent diet composition may influence baseline variation in vaginal microbiota composition, consequently impacting susceptibility or resistance to colonization by microbiota such as GBS. Our results emphasize the importance of benchmarking proper control diets in such studies.

In conclusion, our studies emphasize the significance of diet composition, particularly diets that are high in fat and low in soluble fiber, in shaping vaginal microbiota composition and the response to GBS colonization. A better understanding of these fundamental processes can provide novel insights into approaches that may mitigate the health consequences of ascending infections in the female reproductive tract and related adverse health outcomes.

## MATERIALS AND METHODS

### Animals

All experiments were approved by the University of Pittsburgh and Magee-Womens Research Institute Institutional Animal Care and Use Committee and performed in accordance with National Institutes of Health Animal Care and Use Guidelines. C57Bl/6 N mice were from Taconic Biosciences (C57Bl/6NTac) and arrived in the animal facility aged 3 weeks. Mice were allowed to acclimate 1 week prior to experimentation. All mice were maintained on a 12-h light/dark cycle (lights on: 0600 EST; lights off: 0180 EST. An Onset HOBO MX2202 Wireless Temperature/Light Data Logger (HOBO Data Loggers, Wilmington, NC) was used to confirm stability of light: dark photoperiod. *Ad libitum* access was provided to water and a Chow diet (PicoLab Mouse Diet 20, LabDiet, CA). On average, the Chow diet supplied energy as 20.16% fat, 23% protein, 55% carbohydrates, and 15% dietary fiber (range of total dietary fiber was between 15% and 25% with 15%–20% insoluble and 2%–5% soluble fiber). Two refined diets were used during the experiment: refined low-fat/low soluble fiber diet (rLFD, Research Diets D12450J, ResearchDiets, NJ) supplying energy as 12% fat, 21% protein, and 67% carbohydrates; and refined high-fat/low soluble fiber diet (rHFD, Research Diets D12492; ResearchDiets, NJ) supplying

energy as 45% fat, 20% protein, and 35% carbohydrates. Following acclimation, mice were randomized to either remain on Chow or switched to rHFD or rLFD. Body weight was recorded once per week.

## Glucose tolerance test

Glucose tolerance test was administered in 10-week-old females following 6 weeks on either Chow, rLFD, or rHFD diets. Food was removed at 0800 EST, and mice were fasted for 6 h, upon which mice were injected intraperitoneally with 1 mg/kg BW of 0.3 g/mL of glucose in saline. Glucose readings were collected by tail blood at 0-, 15- to 30-, 60-, and 120-min timepoints using the Contour Next Blood Glucose Monitoring System (Bayer Co, Germany). Previous work revealed that ~15% of females consuming the rHFD diet were resistant to diet-induced body weight gain and intolerance. As a result, we established the following inclusion criteria: females on the respective diets that do not overlap by two standard deviations on body weight and glucose tolerance test were excluded from the study. Using this exclusion criteria, five females consuming rHFD were removed from subsequent analysis, and no rLFD and Chow females were excluded.

## *Streptococcus agalactiae* COH1 colonization and clearance assessment (GBS COH1)

Murine vaginal colonization of nonpregnant C57BL/6 J mice was performed as previously described (43, 44). Eleven-week-old mice were injected subcutaneously with 0.5 mg of water-soluble/cyclodextrin-encapsulated β-estradiol on two successive days to synchronize estrus. On the third day, stationary phase GBS COH1 was pelleted and resuspended in a 1:1 mixture of phosphate-buffered saline (PBS) and a sterile 10% gelatin solution to enhance viscosity. This culture preparation was diluted and plated for CFU enumeration. Mice were vaginally colonized by shallow insertion of the pipette tip into the vagina followed by depression of the pipette plunger to administer 50 μL (corresponding to $1 \times 10^7$ CFU) of culture preparation, then removal of pipette tip from the vagina prior to plunger release. A vehicle group received only 10% gelatin in PBS to compare with females that showed successful colonization. Once the vaginal inoculation procedure was complete, the mouse was placed in a new, clean cage and single housed for the remainder of the study. After 48 h/2 dpi, an initial swab to detect colonization was performed. A Copan Flexible Minitip Flocked Swab (Eswab 482C) was moistened in 500 μL of sterile liquid Amies transport medium, then inserted into the murine vagina and rotated three times, after which it was placed in the Amies transport medium tube and swirled to release adherent GBS. The PBS was diluted and plated for CFU enumeration on GBS-specific Chromagar plates (Chromagar, Paris, France, cat. # SB282), which were then grown overnight at 37°C. Mice with established vaginal colonization, based on the 48-h swab, were then swabbed three times weekly (Monday, Wednesday, and Friday) following the above protocol for a total of 30 days. Colonization clearance was defined as two sequential swabs with no detectable GBS. The remaining content in the tube is then frozen and stored at – 80°C until downstream processing.

## Vaginal fluid DNA extraction and 16S rRNA marker gene sequencing

The MagAttract PowerMicrobiome DNA/RNA Kit (Qiagen) extracted genomic DNA from 200 μl of liquid Amies containing microbial specimen, using bead beating on a TissueLyser II (Qiagen), according to the manufacturer's instructions. 16S libraries were generated using a two-step PCR protocol. The 16S Amplicon PCR Forward Primer was 5′ TC GTC GGC AGC GTC AGA TGT GTA TAA GAG ACA GCC TAC GGG NGG CWG CAG. The 16S Amplicon PCR Reverse Primer was 5′ GTC TCG TGG GCT CGG AGAT GTG TAT AAG AGA CAG GAC TAC HVG GGT ATC TAA TCC (106). Amplicon PCR was performed as follows for amplification of the 16S rRNA V3–V4 region from vaginal swab fluid: initial denaturation at 95°C for 3 min, followed by 25 cycles of 95°C for 30 s, 55°C for 30 s, 72°C for 30 s, and a final extension at 72°C for 5 min. Resultant 16S V3–V4 amplicons were purified

using AMPure XP beads at a 0.8 ratio of beads to amplicon volume. Illumina Nextera XT v2 Index Primer 1 (N7xx) and Nextera v2 XT Index Primer 2 (S5xx) were used as index primers. Index PCR was performed as follows for the amplification of the 16S rRNA V3–V4 region from vaginal fluid lavage: initial denaturation at 95℃ for 3 min, followed by 8 cycles 95℃ for 30 s, 55℃ for 30 s, 72℃ for 30 s, and a final extension at 72℃ for 5 min. Result-indexed libraries were cleaned up using AMPure XP beads at a 1.12 ratio of beads to indexed library. The concentration of indexed libraries was quantified using Qubit, and library fragment size was quantified using an Agilent Tapestation 4200 with D5000 ScreenTapes. Libraries were normalized, pooled, and a paired-end sequencing of pooled libraries was done on an Illumina iSeq 100 System using $2 \times 150$-bp run geometry in our laboratory.

## Processing and analysis of 16 rRNA marker gene sequencing data

The sequences were demultiplexed on the BaseSpace Sequence Hub using the bcl2fastq2 conversion software version 2.2.0. Quality control on the resulting demulti-plexed forward fastq files were performed using DADA2 denoise-single function, and R1 and R2 fastq files were concatenated (107). A Naive Bayes feature classifier was trained using SILVA reference sequences with the q2-feature-classifier for taxonomic analysis. The total read count was 9,449,935, and the average count per sample was 30,385. Data filtering was set to include features where 20% of its values contain a minimum of four counts. In addition, features that exhibit low variance across treatment condi-tions are unlikely to be associated with treatment conditions, and therefore, variance was measured by interquartile range and removed at 10%. Data were normalized using trimmed mean of M-values. For quality control purposes, water and processed blank samples were sequenced and analyzed through the bioinformatics pipeline. Taxa identified in the contamination blanks, cyanobacteria, and "unclassified" to the phylum level were removed (108, 109).

## Quantification and statistical analysis

Statistical information, including sample size, mean, and statistical significance values, is shown in the text or figure legends. A variety of statistical analyses were applied, each one specifically appropriate for the data and hypothesis, using R version 4.1.0 (R Core Team, 2021). ANOVA testing with repeated measures corrections and Tukey *post-hoc* tests were used, with significance at an adjusted $P < 0.05$. GraphPad Prism and Adobe Illustrator were used for generating figures. All analyses were performed in python version 3.6.12 (Python Software, 2020) or R version 4.1.0 (R Core Team, 2021).

## ACKNOWLEDGMENTS

This work is supported by funding from the National Institute of Allergy and Infectious Disease grant R21AI147511 (PI: Hooven), Eunice Kennedy Shriver National Institute of Child Health and Human Development grants T32HD087194 (PI: Munyoki) Pilot Project from P50HD096723 (PI: Jašarević), K12 HD000849 (PI: Megli), the National Institute of Diabetes and Digestive and Kidney Diseases grant K01DK1121734 (PI: Jašarević), a Magee Auxiliary Research Scholars Award (PI: Jašarević) Burroughs Wellcome Fund Next Generation Pregnancy Initiative (PI: Jašarević). We acknowledge Hemanjani Bhavana for the bacterial culture of GBS COH1 isolates, Heather Evers, Cheyenne Miller, and Heather Seiple in the Magee-Womens Research Institute animal facility for the technical assistance in the care of mice. We acknowledge the University of Pittsburgh Center for Research Computing through the resources provided. Specifically, this work used the HTC cluster, which is supported by NIH award number S10OD028483. The funders had no role in the study design, data collection or analysis, decision to publish, or preparation of the manuscript.

C.J.M.: Conceptualization, Methodology, Investigation, Resources, Visualization, Formal analysis, Writing—original draft, reviewing, and editing; A.D.: Conceptualization,

Methodology, Validation, Writing—review and editing; J.P.G.: Conceptualization, Methodology, Validation, Data curation, Supervision, Project administration, Writing—review and editing; S.K.M.: Software, Formal analysis, Data curation, Visualization, Writing—review and editing; T.A.H.: Conceptualization, Methodology, Resources, Supervision, Writing—original draft, reviewing, and editing; E.J.: Conceptualization, Formal analysis, Resources, Visualization, Supervision, Project administration, Funding acquisition, Writing—original draft, reviewing, and editing.

## AUTHOR AFFILIATIONS

[1]Department of Obstetrics, Gynecology and Reproductive Sciences, University of Pittsburgh School of Medicine, Pittsburgh, Pennsylvania, USA

[2]Division of Maternal–Fetal Medicine, UPMC Magee-Womens Hospital, Pittsburgh, Pennsylvania, USA

[3]Division of Reproductive Infectious Disease, UPMC Magee-Womens Hospital, Pittsburgh, Pennsylvania, USA

[4]Magee-Womens Research Institute, Pittsburgh, Pennsylvania, USA

[5]Department of Infectious Diseases and Microbiology, University of Pittsburgh School of Public Health, Pittsburgh, Pennsylvania, USA

[6]Department of Computational and Systems Biology, University of Pittsburgh School of Medicine, Pittsburgh, Pennsylvania, USA

[7]Department of Pediatrics, University of Pittsburgh School of Medicine, Pittsburgh, Pennsylvania, USA

[8]Richard King Mellon Institute for Pediatric Research, University of Pittsburgh Medical Center, Pittsburgh, Pennsylvania, USA

[9]UPMC Children's Hospital of Pittsburgh, Pittsburgh, Pennsylvania, USA

## AUTHOR ORCIDs

Thomas A. Hooven  http://orcid.org/0000-0003-1959-186X

Eldin Jašarević  http://orcid.org/0000-0001-8174-0710

## FUNDING

| Funder | Grant(s) | Author(s) |
| --- | --- | --- |
| Burroughs Wellcome Fund (BWF) | 1050850 | Eldin Jašarević |
| HHS | NIH | National Institute of Diabetes and Digestive and Kidney Diseases (NIDDK) | K01DK1121734 | Eldin Jašarević |
| HHS | NIH | Eunice Kennedy Shriver National Institute of Child Health and Human Development (NICHD) | P50HD096723 | Eldin Jašarević |
| HHS | NIH | National Institute of Allergy and Infectious Diseases (NIAID) | R21AI147511 | Thomas A. Hooven |
| HHS | NIH | Eunice Kennedy Shriver National Institute of Child Health and Human Development (NICHD) | HD000849 | Christina J. Megli |
| HHS | NIH | Eunice Kennedy Shriver National Institute of Child Health and Human Development (NICHD) | T32HD087194 | Sarah K. Munyoki |

## AUTHOR CONTRIBUTIONS

Christina J. Megli, Conceptualization, Formal analysis, Investigation, Methodology, Resources, Writing – original draft, Writing – review and editing | Allison E. DePuyt, Investigation, Methodology, Visualization | Julie P. Goff, Conceptualization, Data curation, Project administration, Writing – review and editing | Sarah K. Munyoki, Formal analysis, Software, Validation, Visualization, Writing – review and editing | Thomas A. Hooven,

Conceptualization, Methodology, Resources, Writing – original draft, Writing – review and editing | Eldin Jašarević, Conceptualization, Data curation, Formal analysis, Funding acquisition, Investigation, Methodology, Project administration, Resources, Visualization, Writing – original draft, Writing – review and editing

## DATA AVAILABILITY

Accession numbers for the raw fastq files have been deposited to NCBI and available under PRJNA1081370.

## ETHICS APPROVAL

All experiments were approved by the University of Pittsburgh and Magee-Womens Research Institute Institutional Animal Care and Use Committee and performed in accordance with National Institutes of Health Animal Care and Use Guidelines.

## ADDITIONAL FILES

The following material is available online.

### Supplemental Material

**Table S1 (Spectrum03623-23-s0001.xlsx).** Showing full ANOVA results.
**Table S2 (Spectrum03623-23-s0002.xlsx).** Showing full ANOVA results.

### Open Peer Review

**PEER REVIEW HISTORY (review-history.pdf).** An accounting of the reviewer comments and feedback.

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
