## [Reviewer comments · Microbiology Spectrum]

Microbiology Spectrum

Diet influences community dynamics following vaginal group B Streptococcus colonization

Christina Megli, Allison DePuyt, Julie Goff, Sarah Munyoki, Thomas Hooven, and Eldin Jasarevic

Corresponding Author(s): Eldin Jasarevic, University of Pittsburgh School of Medicine

Review Timeline:

Submission Date:	October 10, 2023
Editorial Decision:	November 20, 2023
Revision Received:	March 4, 2024
Accepted:	April 22, 2024

Editor: Kevin Theis

Reviewer(s): Disclosure of reviewer identity is with reference to reviewer comments included in decision letter(s). The following individuals involved in review of your submission have agreed to reveal their identity: Kathryn A. Patras (Reviewer #1); Nicole Gilbert (Reviewer #2)

Transaction Report:

DOI: <https://doi.org/10.1128/spectrum.03623-23>

Re: Spectrum03623-23 (Diet influences community dynamics following vaginal group B Streptococcus colonization)

Dear Dr. Eldin Jasarevic:

Thank you for the privilege of reviewing your work. The manuscript has been reviewed by two experts in the field. In general, they are favorable towards the work. Yet, further clarity is requested. Also, please add the Bioproject ID to the Data Availability Statement within the document.

Below you will find instructions from the Spectrum editorial office, and the reviewer comments. Please return the manuscript within 60 days; if you cannot complete the modification within this time period, please contact me. If you do not wish to modify the manuscript and prefer to submit it to another journal, notify me immediately so that the manuscript may be formally withdrawn from consideration by Spectrum.

Revision Guidelines

Sincerely,
Kevin R. Theis
Editor
Microbiology Spectrum

Reviewer #1 (Comments for the Author):

Megli et al present new work on the interplay of diet and metabolic disease on the vaginal microbiota and pathobiont GBS. GBS is an important perinatal pathogen and the maternal vaginal tract is a critical reservoir. Although the vaginal microbiota has been implicated in mediating GBS colonization, specific mechanisms are not well-described. Using a high fat low fiber diet-induced mouse model of pubertal-onset obesity, the authors interrogate the impact of diet and metabolic disease (obesity, glucose

intolerance) on GBS and its interactions with the vaginal microbiota. Two diets were used as controls - standard chow and a macro-nutrient matched low fiber low fat diet. They observed differential rates of GBS colonization over a 30-day period, with both low fiber diets exhibiting sustained GBS colonization. Although markers of metabolic disease were present in the high fat diet group, no correlations with GBS burdens and metabolic dysfunction were observed. As seen in prior studies, the vaginal microbiome became dominated by GBS following inoculation, and six endogenous taxa were impacted by GBS introduction. Of note, *Enterococcus* and *E. coli/Shigella* were increased in the high fat diet group. Overall, this manuscript provides new and important insight into the impact of diet on the vaginal microbiota in a murine model and colonization by GBS. The experimental design is appropriately controlled, meticulously analyzed, and the text is well-written. My comments are geared towards providing additional context and clarity and are all considered minor.

Specific comments:

- The introduction would benefit from a brief comment on clinical associations between obesity and GBS colonization and disease.
- It would be helpful to include more details on vehicle control mice (Fig. 3A): what time points and what diets are represented?
- Were any associations between metabolic parameters and the 16S data observed?
- Figure 3B: are these samples from mice inoculated with GBS, or at baseline? Please clarify.
- Figure 3G and Lines 252-253 and Lines 292-293:: clarify what is meant by counts (OTU? ASV?)
- The discussion would benefit from mentioning a limitation of the inability to distinguish between GBS/diet variables and effects of other confounding variables such as post-pubertal fluctuation and adjustment to new vivarium.
- Line 313: this is an important point. Were *S. agalactiae* reads detected in vehicle treated mice?

(Very) Minor comments:

- Lines 589-591: This sentence references a section that is not present in this manuscript. Additionally, experimental n are not present in figure legends, but should be added.
- In figure legends (Fig. 3-4), almost every panel is referenced as a barplot; however, many of the panels are not barplots.
- Lines 125-126 and line 511-513: Incongruency in referenced diets.
- Lines 523-525: Please indicate the number of mice that met this exclusion criteria.
- Lines 52-53: Bacterial vaginosis is referenced twice.
- Line 166: typo in *agalactiae*.
- Lines 597-598: Please ensure accession numbers for sequence data are made available in the published version.

Reviewer #2 (Comments for the Author):

It appears like this is a straightforward observational study of GBS colonization and vaginal microbiome composition in mice fed different diets, however, it is difficult to discern the methodological and experimental rigor of the study because key information is missing from the results and figure legends. For example:

1. I see Figure 1 legend states the mouse numbers in each group. Why are these not matched? How were the group numbers determined? Were power calculations performed to ensure significant differences could be observed if they were present?
2. Was the GBS colonization experiment performed in a single experiment or multiple independent experiments? If multiple, how many mice per group were in each experimental run and how was inter-experimental variability assessed? I would hesitate to put much stock in any observed differences if all the data come from a single experiment.
3. At what timepoint were the mice estrogenized and inoculated with GBS? It would be much more useful to modify the schematic that is currently Fig 1A so that it includes the GBS inoculation experimental timeline.
4. The experiments only included mice that established vaginal colonization by GBS based on the 48 h swab. Was the rate of positivity at this timepoint the same between all diet groups?
5. Are the data in Figure 3E-G from all samples? Only GBS colonized? Only vehicle?

Other comments:

1. The data in Figure 3 show the expected result that when mice are inoculated with GBS, then GBS becomes the dominant taxa. The authors spend a lot of time talking about shifts in diversity without fully acknowledging that of course diversity will shift when mice are inoculated with a bacterial species that is known to be able to colonize the niche. Of course "GBS colonization affects vaginal microbiota alpha diversity," because now there is a whole new organism in the population. GBS colonization is correlated with differences in vaginal community structure simply because now there is abundant GBS present.
2. The authors, and many other researchers, have already established the weight gain and glucose level phenotypes shown in Figure 1. In this manuscript, these data are just needed to show that the diets had the expected effects. Figure 1 B-F should be moved to supplemental material. and the section describing these data should be substantially shortened.
3. In general, the authors use language that goes beyond the conclusions that can be drawn from their data. For example:

- a. "Together, our studies are designed to address a crucial gap in understanding how diet host-microbe interactions mold the vaginal microbiota..." The data in this paper do not speak at all to how diet affects the microbiome or GBS or interactions between these and the host. This should be rephrased to "our studies are designed to determine whether diet affects the composition of the vaginal microbiome..."
- b. "This study shows that dietary components, such as dietary fat and soluble fiber, impact GBS density..." The authors have not directly or rigorously investigated specific dietary components. This should be rephrased to "This study shows that mice fed diets with different nutritional composition display differences in GBS density and timing of clearance in the female reproductive tract."
- c. "these results shed light on the interactions between diet composition, vaginal microbiota and GBS COH1 colonization dynamics in a mouse model." This paper does not examine interactions so this should be changed to "associations"
4. In the abstract, *Streptococcus agalactiae* should be italicized.
5. The last paragraph of the results section is more appropriate for the discussion section.
6. Why do the authors mention *Gardnerella vaginalis* at the end of the discussion? This seems to come out of nowhere, and any number of vaginal pathogens or pathobionts could have been mentioned. Any particular reason diet would affect *Gardnerella*, per se?

It appears like this is a straightforward observational study of GBS colonization and vaginal microbiome composition in mice fed different diets, however, it is difficult to discern the methodological and experimental rigor of the study because key information is missing from the results and figure legends. For example:

1. I see Figure 1 legend states the mouse numbers in each group. Why are these not matched?
2. Was the GBS colonization experiment performed in a single experiment or multiple independent experiments? If multiple, how many mice per group were in each experimental run and how was inter-experimental variability assessed?
3. At what timepoint were the mice estrogenized and inoculated with GBS? It would be much more useful to modify the schematic that is currently Fig 1A so that it includes the GBS inoculation experimental timeline.
4. The experiments only included mice that established vaginal colonization by GBS based on the 48 h swab. Was the rate of positivity at this timepoint the same between all diet groups?
5. Are the data in Figure 3E-G from all samples? Only GBS colonized? Only vehicle?

Other comments:

1. The data in Figure 3 show the expected result that when mice are inoculated with GBS, then GBS becomes the dominant taxa. The authors spend a lot of time talking about shifts in diversity without fully acknowledging that of course diversity will shift when mice are inoculated with a bacterial species that is known to be able to colonize the niche. Of course “GBS colonization affects vaginal microbiota alpha diversity,” because now there is a whole new organism in the population. GBS colonization is correlated with differences in vaginal community structure simply because now there is abundant GBS present.
2. The authors, and many other researchers, have already established the weight gain and glucose level phenotypes shown in Figure 1. In this manuscript, these data are just needed to show that the diets had the expected effects. Figure 1 B-F should be moved to supplemental material. and the section describing these data should be substantially shortened.
3. In general, the authors use language that goes beyond the conclusions that can be drawn from their data. For example:
 - a. “Together, our studies are designed to address a crucial gap in understanding how diet host-microbe interactions mold the vaginal microbiota...” The data in this paper do not speak at all to *how* diet affects the microbiome or GBS or

interactions between these and the host. This should be rephrased to “our studies are designed to determine whether diet affects the composition of the vaginal microbiome...”

- b. “This study shows that dietary components, such as dietary fat and soluble fiber, impact GBS density...” The authors have not directly or rigorously investigated specific dietary components. This should be rephrased to “This study shows that mice fed diets with different nutritional composition display differences in GBS density and timing of clearance in the female reproductive tract.”
 - c. “these results shed light on the interactions between diet composition, vaginal microbiota and GBS COH1 colonization dynamics in a mouse model.” This paper does not examine *interactions* so this should be changed to “associations”
4. In the abstract, *Streptococcus agalactiae* should be italicized.
 5. The last paragraph of the results section is more appropriate for the discussion section.
 6. Why do the authors mention *Gardnerella vaginalis* at the end of the discussion? This seems to come out of nowhere, and any number of vaginal pathogens or pathobionts could have been mentioned. Any particular reason diet would affect Gardnerella, per se?

We thank the reviewers for their insightful feedback. We have made every effort to address reviewer comments thoroughly and we believe that the revised manuscript is considerably stronger. Our responses to the reviewer comments appear in blue and associated track changes in the manuscript also appear in blue in the marked-up manuscript. A clean copy is also provided.

Reviewer #1:

1. The introduction would benefit from a brief comment on clinical associations between obesity and GBS colonization and disease.

We have added a brief comment on clinical associations between obesity and GBS colonization and disease in the Introduction. This section reads as follows:

“For instance, preconception obesity has been found to be associated with an increased risk of rectovaginal group B streptococcus (GBS) colonization in multiple studies. A retrospective cohort study of women with singleton term pregnancies showed that obese pregnant women were significantly more likely to be colonized by GBS compared to nonobese controls, even after adjusting for other factors such as race, parity, smoking, and diabetes. The prevalence of GBS colonization in the entire cohort was relatively high (25.8%), and obese gravidas had a 35% higher likelihood of testing positive for GBS compared to nonobese women [1]. Another study also supported the association between obesity and GBS colonization, indicating that maternal obesity is a significant risk factor for GBS colonization [2]. Chronic inflammatory changes and differences in intestinal microbiota in individuals with obesity have been suggested as potential contributors to the increased risk of GBS colonization [3].”

2. It would be helpful to include more details on vehicle control mice (Fig. 3A): what time points and what diets are represented?

The Vehicle control mice reflects samples from mice from all dietary groups that were inoculated with gelatin only and vaginal swab collected 48hrs post-inoculation with gelatin only. We have added this additional detail to the figure legend.

3. Were any associations between metabolic parameters and the 16S data observed?

This is an excellent suggestion. In line with the Reviewer's recommendation, we utilized MaAsLin2 to examine potential associations between the 16S data and metabolic parameters and found no significant correlations. This could be due to several factors, including but not limited to the comparison of longitudinal dynamics (vaginal microbiota) with a single timepoint measure (GTT), or perhaps the measures used may not be sufficiently sensitive to detect associations, such as body weight. It is also possible that the lack of association observed between metabolic parameters and vaginal microbiota mirrors findings in gut microbiota studies. Dalby and colleagues (Cell Reports, 2017) found that feeding mice diets either low or high in refined fats led to significant changes in gut microbiota composition compared to a standard chow diet, but increases in body weight, body fat, and glucose intolerance were only observed in mice fed the high-fat diet. While this study was in male mice, we recently confirmed this dietary uncoupling in female mice as well (Morrison et al., Microbiome 2020). The data from our current studies suggest that a similar pattern may be present in the vaginal microbiota, indicating a potential dietary uncoupling from metabolic parameters. However, this hypothesis is speculative, and we are planning to rigorously test this hypothesis in future studies.

We provide a small paragraph in Discussion to address this:

Our inability to detect significant associations between body weight, glucose tolerance, GBS clearance, and vaginal microbiota profiles may be related to several factors. First, the lack of association could arise from the nature of the endpoints being compared, including comparison of longitudinal dynamics (vaginal microbiota) with a single timepoint measure (glucose tolerance test, GTT), or the measures used are not sufficiently sensitive to detect associations, such as

with body weight. Second, the observed lack of association between metabolic parameters and vaginal microbiota may reflect recent findings in gut microbiota studies. For instance, recent work from our group and others showed that feeding mice diets either rLFD or rHFD led to significant changes in gut microbiota composition compared to a standard chow diet (Dalby, Morrison). However, increases in body weight, body fat, and glucose intolerance were only observed in rHFD mice. Our data may suggest that a similar pattern may exist in the vaginal microbiota, indicating a potential dietary uncouple from microbiota effects and metabolic parameters. A rigorous examination of this hypothesis should be the focus of future research.

4. Figure 3B: are these samples from mice inoculated with GBS, or at baseline? Please clarify.

We appreciate the reviewer's attention to this and have provided an updated figure demonstrating the timeline of experiments as a new part of the figures to further clarify the experimental design.

The figure is reproduced below:

5. Figure 3G and Lines 252-253 and Lines 292-293: clarify what is meant by counts (OTU? ASV?)

We have clarified this to be ASV counts and have corrected the text as demonstrated.

6. The discussion would benefit from mentioning a limitation of the inability to distinguish between GBS/diet variables and effects of other confounding variables such as post-pubertal fluctuation and adjustment to new vivarium.

Given that all animals were received together for the entire experiment and acclimated at the same time to the new vivarium, potential variance introduced by adjustment to a new vivarium is equally distributed across all groups, and our endpoints measured likely eclipse any potential confounds related

to vivarium adjustment. Additionally, we have added a paragraph in the discussion focused on the limitation to distinguish between GBS and diet variables for all studies currently assessing the role of GBS colonization and the effects of confounding variables that impact the outcomes current studies on GBS colonization, vaginal microbiota, and health outcome:

“Furthermore, our study has some limitations, including the inability to directly distinguish the effects of post-pubertal hormone signaling, altered immune signaling associated with obesity, vaginal tissue effects of obesity, and the impacts on immune signaling and tissue regulation in impaired glucose tolerance, as well as the mechanisms underlying bacterial colonization in the vagina. We hypothesize that the determinants of GBS colonization and persistence in this model are multifactorial. Future research should aim to elucidate the mechanistic links between these factors and the observed phenotypes.”

-Line 313: this is an important point. Were *S. agalactiae* reads detected in vehicle treated mice?

As shown in Fig. 3H, no *S. agalactiae* reads were detected in vehicle treated mice. We have revised the text to highlight this critical point in the referenced section.

(Very) Minor comments:

-Lines 589-591: This sentence references a section that is not present in this manuscript. Additionally, experimental n are not present in figure legends, but should be added.

We have corrected this error and removed this line. We have included experimental sample numbers in figure legends.

-In figure legends (Fig. 3-4), almost every panel is referenced as a barplot; however, many of the panels are not barplots.

Correct to now to read, Dot plots.

-Lines 125-126 and line 511-513: Incongruity in referenced diets.

This has been corrected. The correct Catalog number for rHFD is D12492 and rLFD is D12450J.

-Lines 523-525: Please indicate the number of mice that met this exclusion criteria.

The number of mice that this exclusion criteria is reflected in sample numbers in our figure legends, and we have added these numbers in Methods.

-Lines 52-53: Bacterial vaginosis is referenced twice.

Corrected.

-Line 166: typo in *agalactiae*.

Corrected.

-Lines 597-598: Please ensure accession numbers for sequence data are made available in the published version.

The accession number for the sequence data is: PRJNA1081370. It has been updated in the manuscript.

Reviewer #2 (Comments for the Author):

1. I see Figure 1 legend states the mouse numbers in each group. Why are these not matched? How were the group numbers determined? Were power calculations performed to ensure significant differences could be observed if they were present?

We have previously shown (Jasarevic et al., 2021 *Nature Communications*) that ~10-20% of females consuming the rHFD are resistant to body weight gain and glucose intolerance. In anticipation of this probability of resistance in this mouse model and to ensure our studies are sufficiently powered, our approach is to increase the number of animals represented in the rHFD. As discussed in *Glucose Tolerance Test*, our inclusion criteria for the response to rHFD-associated metabolic syndrome requires that females on rHFD do not overlap with Chow and rLFD. We have altered the text to reflect this, and a specific reference appears in the main text:

We have previously demonstrated that 10-20% of the mice consuming the rHFD are resistant to weight gain and glucose tolerance (ref), thus we intentionally had this group have twice as many animals in anticipation of an overlapping physiologic response to diet (63).

2. Was the GBS colonization experiment performed in a single experiment or multiple independent experiments? If multiple, how many mice per group were in each experimental run and how was inter-experimental variability assessed? I would hesitate to put much stock in any observed differences if all the data come from a single experiment.

The GBS colonization experiment was performed as one single experiment, based on the large group numbers of mice in this study, as well as the temporal sampling strategy employed in this study. While we agree the rigor would be increased with multiple cohorts, our cultivation-based GBS clearance data matches that of published work using both A909 and COH1 strains in nonpregnant C57Bl/6 mice (e.g., Dammann et al., 2022, *Journal of Infectious Disease*). We have clarified the sample sizes in each experiment in the figure legends.

3. At what timepoint were the mice estrogenized and inoculated with GBS? It would be much more useful to modify the schematic that is currently Fig 1A so that it includes the GBS inoculation experimental timeline.

Following previous work, and as described in *Streptococcus agalactiae COH1 colonization and clearance assessment (GBS COH1)*, mice were estrogenized at 48h and 24h prior to GBS inoculation. We have added in a timeline that includes this to clarify the experimental design:

4. The experiments only included mice that established vaginal colonization by GBS based on the 48 h swab. Was the rate of positivity at this timepoint the same between all diet groups?

The rate of positivity at this timepoint was the same between all diet groups. We have added this information in the main text:

“Quantification of CFU densities showed that initial colonization was not different between groups with 80-90% of groups with CFU detected at 48 hours.”

5. Are the data in Figure 3E-G from all samples? Only GBS colonized? Only vehicle?

Figure 3D and 3H shows the comparison of vehicle and GBS colonized mice. The remainder the figures includes samples from GBS colonized animals. This is now reflected in the figure legend.

Other comments:

1. The data in Figure 3 show the expected result that when mice are inoculated with GBS, then GBS becomes the dominant taxa. The authors spend a lot of time talking about shifts in diversity without fully acknowledging that of course diversity will shift when mice are inoculated with a bacterial species that is known to be able to colonize the niche. Of course, "GBS colonization affects vaginal microbiota alpha diversity," because now there is a whole new organism in the population. GBS colonization is correlated with differences in vaginal community structure simply because now there is abundant GBS present.

We agree that the introduction of a competitive bacterial species such as GBS into a niche will inherently alter microbial diversity. This point has been further clarified in the revised manuscript to ensure it is adequately addressed. We appreciate the Reviewer highlighting the predictability of these shifts, as it underscores the reliability and robustness of our experimental design and findings. We also wish to emphasize an important aspect of our results that extends beyond the anticipated impact of GBS colonization on microbial diversity. Our data reveal that dietary factors, specifically a low-fiber diet, can significantly influence the composition of the vaginal microbiota and the dynamics of GBS clearance. This finding adds a novel dimension to our understanding of the interplay between diet, microbiota, and pathogen colonization. We believe this adds substantial value to our study by linking dietary habits to microbial and pathogen dynamics in the cervicovaginal space.

2. The authors, and many other researchers, have already established the weight gain and glucose level phenotypes shown in Figure 1. In this manuscript, these data are just needed to show that the diets had the expected effects. Figure 1 B-F should be moved to supplemental material, and the section describing these data should be substantially shortened.

We appreciate the reviewer's suggestion to move Figures 1 B-F to the supplemental material and understand the concern for brevity and focus on the main text. While these data might appear confirmatory, their inclusion in the main text is critical for two reasons. First, they establish a direct link between diet exposure and their physiological impacts, providing essential context no matter how confirmatory. We believe this ensures the narrative's flow and coherence, allowing readers to access foundational information without diverting to supplemental material. Second, differences in outcomes across studies necessitates a clear presentation of the specific effects in our study.

We recognize the reviewer's concerns regarding the length and focus of the manuscript, we have streamlined the section describing these data. The associated text now reads:

We first validated diet effects on body composition using a model for obesity and glucose intolerance that starts during puberty in female C57BL/6N-TAC mice (63) (**Fig. 1A**). Initially, all mice were fed a standard Chow diet from birth until weaning. At weaning, on postnatal day 28, the mice were randomly placed on one of three diets: a low-fat, low-soluble fiber diet (rLFD, Research Diets D12421); a high-fat, low-soluble fiber diet (rHFD, Research Diets D12451); or remained on the Chow diet (**Fig. 1A,B**). We have previously demonstrated that 10-20% of the mice consuming the rHFD are resistant to weight gain and glucose tolerance (ref), thus we intentionally had this group have twice as many animals in anticipation of an overlapping physiologic response to diet (63).

Upon switching to these diets, significant differences in body weight trajectories were observed, consistent with prior findings (63, 64). Our current analysis showed significant effects of both time and diet, as well as their interaction, indicating that diet-related weight changes accumulated over time (repeated measures ANOVA, main effect of time: $F_{5,280} = 218.3$, $P < 0.0001$; main effect of diet: $F_{2,56} = 14.05$, $P < 0.0001$; interaction: $F_{10,280} = 11.47$, $P < 0.0001$). Specifically, mice on the rHFD gained weight more rapidly than those on the Chow or rLFD diets, while weight gain trajectories for Chow and rLFD diets were similar (**Fig. 1C**). Six weeks after switching on respective diets, we assessed the impact on glucose tolerance using an intraperitoneal glucose tolerance test. We found that changes in glucose tolerance were linked to time, diet, and their interaction (repeated measures ANOVA, main effect of time: $F_{4,228} = 35.46$, $P < 0.0001$; diet effect: $F_{2,57} = 48.36$, $P < 0.0001$; interaction: $F_{8,228} = 20.48$, $P < 0.0001$). Mice on the rHFD showed a significant delay in glucose clearance compared to those on the Chow and rLFD diets, which had similar glucose clearance rates) (**Fig. 1D**). Additionally, rHFD mice had higher glucose levels after the glucose tolerance test compared to the other groups, while Chow and rLFD mice showed no difference in glucose levels (**Fig. 1D**). We found a significant correlation between weight gain and glucose levels in the rHFD group, but not in the Chow or rLFD groups (rHFD correlation: $r^2 = 0.4004$, $P = 0.0155$; no significant correlations for Chow and rLFD) (**Fig. 1F**). These findings support existing research on how diets with different

fat and soluble fiber contents affect weight gain and glucose tolerance in a diet-specific manner (54, 65, 68, 69).”

3. In general, the authors use language that goes beyond the conclusions that can be drawn from their data. For example:

a. "Together, our studies are designed to address a crucial gap in understanding how diet host-microbe interactions mold the vaginal microbiota..." The data in this paper do not speak at all to how diet affects the microbiome or GBS or interactions between these and the host. This should be rephrased to "our studies are designed to determine whether diet affects the composition of the vaginal microbiome..."

We appreciate the feedback and agree with the reviewer's concerns we have revised the text accordingly.

b. "This study shows that dietary components, such as dietary fat and soluble fiber, impact GBS density..." The authors have not directly or rigorously investigated specific dietary components. This should be rephrased to "This study shows that mice fed diets with different nutritional composition display differences in GBS density and timing of clearance in the female reproductive tract."

We have rephrased this sentence to read, "This study shows that mice fed diets with different nutritional composition display differences in GBS density and timing of clearance in the female reproductive tract."

c. "these results shed light on the interactions between diet composition, vaginal microbiota and GBS COH1 colonization dynamics in a mouse model." This paper does not examine interactions so this should be changed to "associations"

We have changed *interactions* to *associations*.

4. In the abstract, *Streptococcus agalactiae* should be italicized.

We have italicized *Streptococcus agalactiae* in the abstract.

5. The last paragraph of the results section is more appropriate for the discussion section.

We have moved the last paragraph of the results section to Discussion.

6. Why do the authors mention *Gardnerella vaginalis* at the end of the discussion? This seems to come out of nowhere, and any number of vaginal pathogens or pathobionts could have been mentioned. Any particular reason diet would affect *Gardnerella*, per se?

We have removed this passage from the discussion.

Re: Spectrum03623-23R1 (Diet influences community dynamics following vaginal group B Streptococcus colonization)

Dear Dr. Eldin Jasarevic:

Thank you for submitting your work to Microbiology Spectrum. Your manuscript has been accepted, and I am forwarding it to the ASM production staff for publication. Your paper will first be checked to make sure all elements meet the technical requirements. ASM staff will contact you if anything needs to be revised before copyediting and production can begin. Otherwise, you will be notified when your proofs are ready to be viewed.

Sincerely,
Kevin Theis
Editor
Microbiology Spectrum